# Spatio-Temporal Variation in the Exceedance of Enterococci in Lake Burley Griffin: An Analysis of 16 Years’ Recreational Water Quality Monitoring Data

**DOI:** 10.3390/ijerph21050579

**Published:** 2024-05-01

**Authors:** Ripon Kumar Adhikary, Danswell Starrs, David Wright, Barry Croke, Kathryn Glass, Aparna Lal

**Affiliations:** 1National Centre for Epidemiology and Population Health, Australian National University, Canberra 2601, Australia; kathryn.glass@anu.edu.au (K.G.); aparna.lal@anu.edu.au (A.L.); 2Department of Fisheries and Marine Bioscience, Jashore University of Science and Technology, Jashore 7408, Bangladesh; 3Environment, Planning and Sustainable Development Directorate, ACT Government, Canberra 2601, Australia; danswell.starrs@act.gov.au; 4Research School of Biology, Australian National University, Canberra 2601, Australia; 5Lake and Dam, National Capital Authority, Canberra 2601, Australia; david.wright@nca.gov.au; 6Institute for Water Futures, Mathematical Sciences Institute and Fenner School of Environment and Society, Australian National University, Canberra 2601, Australia; barry.croke@anu.edu.au

**Keywords:** urban lake, recreation, health risks, seasonal, enterococci concentration

## Abstract

Recreational waterbodies with high levels of faecal indicator bacteria (FIB) pose health risks and are an ongoing challenge for urban-lake managers. Lake Burley Griffin (LBG) in the Australian Capital city of Canberra is a popular site for water-based recreation, but analyses of seasonal and long-term patterns in enterococci that exceed alert levels (>200 CFU per 100 mL, leading to site closures) are lacking. This study analysed enterococci concentrations from seven recreational sites from 2001–2021 to examine spatial and temporal patterns in exceedances during the swimming season (October–April), when exposure is highest. The enterococci concentrations varied significantly across sites and in the summer months. The frequency of the exceedances was higher in the 2009–2015 period than in the 2001–2005 and 2015–2021 periods. The odds of alert-level concentrations were greater in November, December, and February compared to October. The odds of exceedance were higher at the Weston Park East site (swimming beach) and lower at the Ferry Terminal and Weston Park West site compared to the East Basin site. This preliminary examination highlights the need for site-specific assessments of environmental and management-related factors that may impact the public health risks of using the lake, such as inflows, turbidity, and climatic conditions. The insights from this study confirm the need for targeted monitoring efforts during high-risk months and at specific sites. The study also advocates for implementing measures to minimise faecal pollution at its sources.

## 1. Introduction

Aquatic ecosystems are fundamentally important for human well-being, health, livelihood, and survival [1,2]. While water-based recreation is globally popular [3,4], such exposure can pose direct and indirect risks to human health [5,6]. Elevated levels of faecal bacteria in waterways indicate the possible presence of pathogenic (disease-causing) organisms that also live in human and animal digestive systems (enteric), exposure to which can cause human illness [7,8,9,10].

Globally, the most commonly tested faecal indicator bacteria (FIB) [11] are total coliforms, faecal coliforms, *Escherichia coli*, faecal streptococci, and enterococci [12,13]. The variation of FIB in waterbodies is a result of dynamic relationships among environmental factors, e.g., sunlight, salinity, temperature, disinfection practices, predation, and major sources and sinks, e.g., soil, sediment, beach sand, vegetation, and water types [14]. In addition to these physical and chemical drivers, location-specific factors, such as inflows from surface runoff, high user density, and urbanisation may increase the risk for recreational users, raising concerns for water managers [13,15].

In urban lakes, there are two main sources of indicator bacteria: external and internal [16,17]. Streams and storm drains are major external sources of faecal bacteria and nutrients entering lakes and streams [18,19]. Human enteric pathogens may enter stormwater and, ultimately, surface water through leaking sewage systems, sewer pumping station overflows, and the release of treated wastewater into aquatic habitats [20]. During intense rainfall, bird and animal droppings are also washed into streams and storm drains [21]. Additionally, another key mechanism of faecal bacteria introduction into large water bodies is their adsorption and desorption to sediments, through which they can be transported and released back into water columns over time [22,23]. 

Additional external sources of enterococci are forms of primary-contact recreation, such as swimming and bathing [24,25]. Bathing increases sediment resuspension and redistributes enterococci through the water column. Bathers can also shed FIB. Adult rates of enterococci shedding can range from 1.8 × 10^4^ colony-forming units (CFU) to 2.8 × 10^6^ CFU per bather [26]. The ‘Mirror Lake Jump’ event in Ohio, USA was associated with enterococci densities ranging from 60 CFU per 100 mL up to 9.4 × 10^3^ CFU per 100 mL during the 4-hour peak jumping period [27].

Internally, higher levels of turbidity and suspended solids (e.g., algae, sediment, silt, etc.) are associated with higher FIB concentrations [27,28,29]. As particles provide a more favourable environment than the surrounding water due to increased nutrient or solute availability, physical stability, and protection from predation or other stressors, like chemical disinfectants, particle association is thought to prolong FIB persistence [30]. Particle-associated FIB is also likely to have higher sinking rates, which might affect horizontal movement and, ultimately, contribute to sediment deposition, particularly in shallow water [31]. Wave movement and water flow may result in the re-suspension of settled FIB [32].

Lake Burley Griffin (LBG), in the landlocked city of Canberra, is a place of national importance in terms of creative, technological, and aesthetic heritage [33]. An important part of Canberra’s identity, the lake has become a hub for recreational activities and a place for community cohesion [34]. Elevated levels of FIB leading to alerts, warnings, or beach closures for public health safety disrupt these recreational uses [35].

The water quality of LBG has been impacted by the surrounding catchment through nutrient inputs and sewerage overflows (faecal matter) [36,37,38,39,40,41,42,43], mine-waste pollution [44,45], sediment transport [46], pharmaceuticals and personal care products [47], and allergenic airborne pollen [48]. Further efforts may be necessary to ensure the lake remains safe for recreational activities, despite the significant progress made in identifying catchment-scale contributing factors. By analysing 17 years of historical water-quality data, the National Capital Authority (NCA) Water Quality Plan of 2006 reported that the level of faecal coliform bacteria occasionally exceeded the guideline values [49], especially during late summer and early autumn, and that the upstream concentrations were greater than those at downstream sites in LBG [50]. 

The identification of pollution sources in large bodies of water, such as LBG, is crucial for implementing targeted interventions to mitigate the associated health risks. To achieve this, it is imperative to pinpoint hotspots of bacterial exceedances on beaches. This enables authorities to prioritise remedial actions and allocate resources efficiently to effectively track down and address the sources of contamination. Moreover, analysing temporal patterns provides valuable insights into seasonal fluctuations and the potential drivers of variations in bacterial levels. Furthermore, comparing differences between swimming and non-swimming areas aids in understanding whether specific recreational activities contribute to bacterial pollution, thus informing tailored mitigation strategies for each site’s usage. Therefore, this study seeks to investigate the spatial and temporal patterns of enterococci concentration exceedances across both swimming and non-swimming sites. The goal is to establish a foundational understanding that can inform site-specific management strategies aimed at safeguarding public health.

## 2. Materials and Methods

### 2.1. Study Area

The study was conducted at 7 recreational sites in LBG, together with its inflow catchment (Figure 1). LBG was formed by the damming of the Molonglo River by Scrivener Dam, which was constructed in 1963. Outflows from Scrivener Dam flow down the Molonglo River and, eventually, into the Murrumbidgee River. The LBG catchment area is 1860 km^2^, which includes the Queanbeyan River catchment (960 km^2^), the Molonglo River catchment (780 km^2^), the Jerrabomberra Creek catchment (128 km^2^), and the Sullivan’s Creek catchment (53 km^2^) [48]. These broader LBG catchments are composed of a mix of conservation and recreation lands (27%), urban and intensive lands (5%), and rural lands (68%). Although the proportion of the urban area in the broader LBG catchment is much lower (5%) than that of the urban area in the surrounding LBG catchment (44%), urbanisation and the Australian Capital Territory (ACT)’s regional population are rapidly increasing [51,52]. LBG is administered and monitored by the NCA, which aims to sustain current high-quality water for population health through safeguarding environmental and recreational values [53].

### 2.2. Water Quality Data

We used historical water-quality data from 2001 to 2021 (water-years) on enterococci concentrations collected by the NCA [53] (Table 1). We only used routinely (weekly) collected data during swimming seasons (October to April) (2924 time points), excluding data recorded through resampling (331 time points) or sampling in non-swimming seasons (May to September) (61 time points). There were no reported data from 2006 to 2009. 

### 2.3. Sampling and Laboratory Analysis

Weekly water samples from LBG were collected from each recreational site (Figure 1) in the morning (08:00 a.m. to 12:00 p.m.). Samples were collected aseptically in sterile bottles, placed on ice in a large portable insulated cooler, transported to the laboratory (ALS Global-Testing & Analysis Laboratory, Canberra, Australia) within 3 h, and processed within 6 h of arrival in the laboratory. From 2001 to 2012, the membrane filtration method (USEPA Method 1600) was employed for enterococci analysis. After filtering the water sample through a membrane to capture the bacteria, the membrane containing the bacterial cells was transferred onto a selective medium known as mEI agar. Subsequently, it was incubated for 24 to 48 h at a temperature of 41 °C. During this incubation period, all colonies showing a blue halo, regardless of their color, were identified and recorded as enterococci colonies. To ensure optimal visibility, magnification and a small fluorescent lamp were employed for counting, maximising the clarity and visibility of the colonies [55,56].

From 2012 onward, the Enterolert method, otherwise known as the defined substrate technology (DST) method, was adopted as the preferred approach for enterococci analysis. According to the Enterolert method, (IDEXX Laboratories, Westbrook, ME, USA) [57], in a sterile 100-millilitre bottle, a 1 in 10 dilution of the water sample was prepared (10 mL of sample with 90 mL of sterile distilled water). After shaking to dissolve the powder, one packet of powdered Enterolert reagent was added to the bottle, and the liquid was aseptically poured into a sterile 51-well Quanti-Tray. The tray was then mechanically sealed in a Quanti-Tray sealer, after which the mixture was distributed into the wells simultaneously and incubated at 41 ± 0.5 °C for 24 h. Following incubation, the tray was examined in a darkened room by placing it directly beneath and within 12 cm of a 365-nanometre UV light. Blue fluorescence in a well was regarded as a positive reaction and indicated the presence of enterococci in that well. The number of enterococci per 100 mL was obtained by referring to a 51-well MPN table and multiplying the result by a dilution factor of 10 [58,59]. 

### 2.4. Data Analysis

Statistical analyses were performed using the software R (R version 4.0.4, 15 February 2021, available online: https://www.r-project.org/about.html) [60]. First, descriptive statistics were used to show the distribution of enterococci concentration across recreational sites, seasons, time periods, and by primary use (swimming and non-swimming activities) using the median and quartiles. Two-way tables were constructed to inspect the relationship between dependent enterococci concentration and independent variables, and tested using the Kruskal–Wallis H test.

### 2.5. Enterococci Exceedances

For public health, the threshold value of enterococci in the ACT is ≤200 CFU per 100 mL. If the level of enterococci in a water sample is above 200 CFU per 100 mL, another sample is taken. If the second sample is also above 200 CFU per 100 mL, the swimming area is closed. To reopen it, two consecutive water samples must show that the level of bacteria is below 200 CFU per 100 mL [35,61]. Using this threshold value, all weekly samples > 200 CFU per 100 mL were categorised as an ‘exceedance’. The number and percentage of exceedances for the seven sites across three time periods (2001–2002 to 2004–2005, 2009–2010 to 2014–2015, and 2015–2016 to 2020–2021) were calculated. 

Multi-collinearity was checked using Crammer’s V statistics and the corresponding Chi-square *p*-value. Crammer’s V statistics above 0.5 and the corresponding Chi-square *p* < 0.05 were considered to indicate highly collinear variable pairs, and these were avoided in the model building process [62]. Multivariate logistic regression was used to examine variables that were associated with exceedances of enterococci, with ‘1’ denoting an exceedance (a value of over 200 CFU per 100 mL). We used a purposeful selection of variables [63] to determine the final model. Variables significantly associated with exceedances of enterococci in the univariate test (*p* < 0.05) were selected for the multivariate analysis. Statistical significance for all the analyses in this study was at the 5% level.

## 3. Results

### 3.1. Distribution of Enterococci

The distribution of the enterococci concentrations across the spatial and temporal variables is presented in Figure 2, Appendix A. A Kruskal–Wallis H test showed that there was statistically significant (*p* < 0.05) variation in the enterococci concentrations across the sites, months, years, periods, and primary uses of the sites. The median enterococci concentration was 60 CFU per 100 mL lake water, at the Weston Park East site, the highest among the seven recreational sites. The lowest median concentration was 14 CFU per 100 mL, at the Ferry Terminal site.

Over the period studied, November, December, and February had the highest median enterococci concentration, 40 CFU per 100 mL, followed by January, in which the median enterococci concentration was 36 CFU per 100 mL. October and April had relatively low median enterococci concentrations of 16 and 22 CFU per 100 mL, respectively. The median enterococci concentration was higher (40 CFU per 100 mL) in the designated swimming sites than in the areas without primary-contact recreation (median enterococci 24 CFU per 100 mL).

### 3.2. Exceedance

Figure 3 and Appendix A present the distribution of the enterococci concentration exceedances across the different sites, months, water-years, periods, and primary uses of LBG sites. There were significant differences (*p* < 0.001) in the enterococci exceedances across all the studied variables. Among the sampled sites, Weston Park East exhibited the highest percentage of exceedances, with 18.51% of its samples surpassing the alert threshold, followed by Yarralumla Beach, with exceedances of 13.22%. Conversely, Ferry Terminal had the lowest percentage of exceedances, with only 12.41% of its samples exceeding the threshold.

In terms of months, November had the highest percentage of exceedances (16.74%), followed by February (13.57%), December (11.70%), and January (11.53%). April had the lowest percentage of exceedances, with only 6.49% of samples surpassing the enterococci threshold. Considering the water-year, the highest percentage of samples exceeding the enterococci threshold was observed in the year 2010–2011 (25.62%), followed by 2011–2012 (17.14%) and 2012–2013 (11.11%). On the other hand, the years 2003–2004 (4.76%) and 2018–2019 (5.82%) exhibited lower percentages of exceedance samples, indicating relatively good water quality during these periods.

The findings underscore the variations in the enterococci concentrations across different time periods, with the period from 2009 to 2015 showing the highest percentage of exceedances (14.68%), followed by 2015–2021 (9.31%) and 2001–2005 (8.61%). Over the entire period, the percentage of enterococci concentrations exceeding the recommended public health limit of 200 CFU per 100 mL was higher at recreational swimming sites (13.77%), compared to non-swimming sites (9.19%). 

The exceedances in the enterococci concentrations across the sites over the three periods are presented in Figure 4a and Appendix A. During the period of 2001–2005, Weston Park West had the lowest exceedance proportion, at 2.04%, while Weston Park East had the highest proportion, at 14.43%. In the subsequent period of 2009–2015, all seven sites experienced an increase in exceedance proportions, with Lotus Bay, Yarralumla Beach, and Weston Park West showing more than a two-fold increase. However, Ferry Terminal had a lower proportion, of 8.39%. In the most recent period, of 2015–2021, there was a decrease in the enterococci exceedances across all the sites compared to the previous period. Among the swimming sites, Black Mt. Beach had the lowest exceedance proportion, at 4.27%, in the recent period. Throughout all three periods, Weston Park East consistently had the highest exceedance among the seven sites, peaking at 23.23% in 2009–2015. 

Appendix A reveals notable site-specific and monthly patterns in enterococci exceedances within Lake Burley Griffin. Weston Park East consistently had the highest number of exceedances, particularly in December (17) and November (15), followed by Yarralumla Beach, with a peak in November (18). Examining the results across months, November emerged as the month with the highest exceedance counts across multiple sites, including Yarralumla Beach (18), Weston Park East (15), and Lotus Bay (13). December and January also showed notable exceedance counts, suggesting potential seasonal variations in the enterococci levels. 

Figure 4b also showed variations in the enterococci exceedances across different water-years, with higher counts observed in certain years for specific sites, such as Lotus Bay in 2010–2011 and Weston Park East in 2011–2012. Weston Park West had consistently lower exceedance counts throughout the years.

### 3.3. Collinearity Diagnostics

The collinearity diagnostic matrix identified that the ‘primary use’ variable was highly collinear with the ‘site’ variable. Based on the focus of this study, we did not use primary use in the multivariate model. 

### 3.4. Multivariate Analysis

According to the fitted model (Table 2), holding the year and site constant, the odds of enterococci exceedance (>200 CFU per 100 mL) were 2.38, 1.61, and 1.76 times higher in November, December, and February, respectively (CI = 1.54–3.77, *p* < 0.001; CI = 1.02–2.58, *p* = 0.045 and CI = 1.12–2.84, *p* = 0.018), than the odds in October. When compared to the East Basin site, the Weston Park East site had significantly higher odds of enterococci exceedance (OR = 1.52, 95% CI = 1.04–2.34, *p* = 0.031), while the Ferry Terminal and Weston Park West sites had significantly lower odds (OR = 0.39, 95%, CI = 0.23–0.64, *p* < 0.001 and OR = 0.46, 95% CI = 0.28–0.74, *p* = 0.002), respectively.

## 4. Discussion

The concentration of faecal indicator bacteria, as measured by enterococci bacteria, varied across the sites in LBG, with the recreational swimming sites recording higher median concentrations across the summer months. Weston Park East, Lotus Bay, Yarralumla Beach, and Black Mountain Beach had higher median bacteria concentrations than the other upstream recreational sites. These sites are in the West Lake and West Basin, a popular area for recreational activities such as swimming, boating, and fishing. Such intensive human activities, along with the surrounding land use, could be contributing to higher levels of contaminants in the water. Apart from human activities, the surface water runoff through Sullivans Creek, which is located upstream of these sites (excluding Lotus Bay), may also contribute to the higher bacterial concentration. Sullivans Creek is a significant source of nutrients, such as nitrogen and phosphorus, for LBG [34,64], which may eventually promote the growth of enterococci both in inflows and in receiving water [65]. Urban lakes are greatly influenced by stormwater drainage because of runoff. LBG, as a drainage lake, is influenced by inlet channels or seepage from the surrounding urban catchments [66,67]. These nonpoint sources are likely to contain human and animal faeces, although swimmers and aquatic animals are also likely sources of contamination. It would be beneficial to include monitoring points for faecal bacteria at all major stream discharge locations to better understand the sources and dynamics of faecal bacteria inputs into the lake. Currently, the ACT Government monitors two sites along the Molonglo River (Molonglo Reach and a water-ski area) near the inlet of LBG. Additionally, the implementation of microbial source tracking methodologies is a promising avenue for pinpointing the specific origins of the heightened bacterial concentrations observed in recreational lake and beach environments [68,69].

The Weston Park East swimming site is situated at the edge of Weston Park, one of Canberra’s busiest parks [70]. A higher number of park or beach users could lead to an increase in the overall pressure on the beach infrastructure, including facilities such as public toilets, showers, and waste-disposal facilities [71]. Thus, the excessive accumulation of human waste and other pollutants could contribute to higher levels of faecal bacteria in the water. Increased beach usage can also result in higher levels of litter, which can contribute to the overall degradation of the beach environment and negatively impact water quality. The presence of pet animals, such as dogs, can also contribute to the accumulation of animal waste in the park. Although this study did not focus on these factors, in-depth sanitary inspection might identify site-specific factors that can be controlled through improved management measures to safeguard users’ risk of illness [72]. It may be useful to track visitor numbers and traffic to the sites during the times when these areas are open for swimming. The World Health Organization (2021) recommends such ‘system assessment’ as a vital component of recreational water safety planning (RWSP) [13]. The sites with higher percentages of bacterial concentration that exceeds the alert-level threshold may be reference points for developing further RWSP for better risk assessment and communication. As an immediate measure, the lake authority should consider implementing robust public awareness campaigns aimed at fostering eco-consciousness and encouraging responsible practices for maintaining beach water quality [73]. 

In addition to nonpoint sources of pollution and management-related factors, the hydro-geomorphological and limnological characteristics of this artificial lake, such as the water flow, depth, slopes, types of soil, sand or sediment, the presence of aquatic plants, and debris, among many others, may have contributed to the reported enterococci concentrations at certain sites [74]. According to an LBG investigation report (2012), higher nutrient intake, followed by increased planktonic development and subsequent breakdown, may affect the overall ecology [34] and bacterial in-lake regeneration process, leading to higher levels of bacteria. These potential causes are supported by existing Australian [75] and global research [76,77,78,79,80], which advocate for integrating long-term ecological restoration initiatives into lake management strategies to address faecal bacterial exceedances and foster resilient ecosystems. For instance, riparian restoration efforts, including the establishment of vegetative buffers along the shoreline and soil stabilisation, play a crucial role in filtering pollutants from runoff, thus preventing their entry into the water [81]. Moreover, the creation of diverse habitats through restoration endeavours fosters ecological equilibrium and supports beneficial microorganisms that compete with faecal bacteria [82]. Additionally, measures such as bank stabilisation and water temperature regulation further contribute to improving aquatic health, consequently enhancing water quality and minimising the risk of bacterial contamination [83]. 

The sites with the higher bacterial concentrations were in and around the Tarcoola Reach, which is the downstream region of the lake. The eastern basin of LBG has an important function as a retention system for water quality control. When inflow water from upstream sources such as the Molonglo River or Jerrabomberra Creek enters the lake, it loses its energy and settles in the eastern basin. This settling process serves as a major point of sediment deposition, allowing the lake to maintain its water quality and prevent further pollution downstream [34]. While the western basin of the lake receives a reduced volume of sediments due to the sedimentation that occurs in the eastern basin, it is still possible that sediment deposition could occur over time. The cumulative deposition of sediments or nutrient accumulation in the western basin over the years could create environmental conditions in deeper parts of the downstream lake that are conducive to higher levels of faecal bacteria in the water [36]. LBG’s water quality has been a longstanding concern for stakeholders, prompting strategic planning [84] to address the issue. This planning involves a range of approaches, including in-lake management, urban catchment management, and the management of sewage inputs (both municipal and those distributed from rural catchments) and river flow [34]. The implementation of some of the plans for managing water quality in LBG has been reported [51], while evaluation reports on completed projects are still pending [85]. This study reported a stable or slightly downward pattern in the concentration of enterococci as a measure of the enterococci in recent years, which may reflect the ongoing management interventions implemented by the relevant lake authorities. The authority should prioritise the evaluation of completed projects to assess their effectiveness and identify areas for improvement [86]. 

From November to February, the average concentration of enterococci was elevated across all the sites, across all the years studied. Rain events may have led to an increase in surface runoff and accompanying water flow for the lake [34,87]. According to the summary statistics of the historical rainfall data (2009–2022) recorded at Canberra Airport (station no. 70351, Bureau of Meteorology), the months of November to March had the highest median rainfall (ranging from 54.6 mm to 76.2 mm) compared to the other months [88]. The impact of rainfall was particularly noticeable in the significant increase in enterococci exceedances during the 2010–2011 water-year. During this period, the stream flow in the waterways originating within the ACT surged due to heavy rainfall, leading to flooding in October, November, December, and February. The data for the 12-month reporting period (July 2010–June 2011) revealed that the urban-area rainfall exceeded that of the previous year by 254 mm and that, at 866 mm, it was well above the long-term average [89]. Temperature gradients, in addition to rainfall, also influence the water chemistry, which has a synergistic effect on enterococci concentrations in particular months [14]. While identifying and controlling hydro-meteorological factors may require long-term management measures [90], the NCA has implemented several short-term approaches to reduce health risks for users. For instance, public awareness campaigns are conducted during the summer recreational months to educate beachgoers about the potential health hazards associated with elevated enterococci levels and to encourage them to take necessary precautions. Additionally, the NCA provides information about the latest water quality conditions through on-site signages and various online platforms, including the free Swim Guide App, the Swim Guide website, and the NCA website [91]. This enables beach visitors to make informed decisions regarding primary-contact activities. However, increasing the frequency of water quality monitoring, employing rapid microbial detection technologies [92,93], or providing the real-time communication of beach advisories [94] during high-risk months, especially at vulnerable sites, could enhance safety for recreational water users.

The frequency of enterococci exceedances determines the probability of beach closure days, particularly for the safety of primary-contact activities [95,96]. Frequent beach closures result in economic losses [34,97], as well as a negative public perceptions of both water quality and water-based recreation [98]. According to the WHO guidelines, there are opportunities to revise the exceedance threshold or USEPA Beach Action Value (BAV) [99], the concentration of faecal indicator bacteria in water based on which local beach closure decisions are made [13]. For beach closures, the state and/or local government decides that water conditions are unsafe for swimmers and other users. However, epidemiological evidence is needed to adapt new thresholds for beach closure policies for local or regional water bodies. To optimise beach management and mitigate the negative impacts of unnecessary closures on the local economy and user satisfaction, it is crucial to explore the potential of existing data in forecasting beach status [100]. While long-term water quality monitoring data provide a foundation, leveraging additional existing datasets, including hydrological and meteorological factors, is essential [94,101]. By using this available information, particularly regarding water flow, wave characteristics, temperature, rainfall, air temperature, and wind patterns, a data-driven framework can be trialed for predicting bacteria levels in beach water several days in advance. This approach would streamline decision-making processes and facilitate proactive sampling to validate closure decisions [102].

The observation of clear spatio-temporal variation in the enterococci levels in the upstream lake (East Basin) supports the idea that the water quality of the urban lake is potentially driven by the upstream water catchment. For example, the East Basin showed a marked decline in exceedances in the 2015–2021 period. This site is situated in an upstream corner of the lake, and it is less likely to be influenced by downstream sites. Sudden elevations in enterococci in the East Basin may be due to incoming flow from Molonglo Reach, near the Jerrabomberra Wetlands, faecal runoff from nearby grazing lands, or the Kingston foreshore residential area [34]. Management interventions in recent years, such as the macrophyte growth project, and sewerage system management to reduce the potential catchment pollution in the upper Queanbeyan River, upper Molonglo River, and Jerrabomberra Creek may have contributed to the lower number of exceedances in 2015–2021. Another aspect to note is that our 16-year study included two water-years (2019–2020, 2020–2021) during which the ACT experienced COVID-19-pandemic lockdowns. Despite expectations of decreased enterococci exceedances due to potentially restricted lake usage during lockdowns [103], our data showed similar exceedance patterns to non-pandemic years. Notably, the median enterococci concentration in 2020–2021 was higher than in the preceding two water-years, suggesting that source water quality might better explain this temporal variation [104]. While our findings cannot clearly demonstrate any potential impacts of the COVID-19 lockdowns on the enterococci exceedances in LBG, further research is warranted to comprehensively explore this association.

Overall, this study highlights the importance of addressing both local management practices within urban lakes and factors contributing to contamination in the upstream water catchment to improve water quality. For example, implementing riparian exclosure fencing [105], especially in grazing areas along upstream channels, can restrict animal access to waterways, thus reducing the risk of the direct contamination of water sources with animal waste [106,107]. Vegetative buffers and constructed wetlands are effective interventions for reducing faecal pollution in downstream lakes, but careful management measures are necessary to mitigate the risk of contamination from inhabiting animals and birds [108,109]. The regular maintenance of constructed wetlands, including the removal of accumulated organic matter and periodic cleaning, can prevent the buildup of pollutants. 

## 5. Conclusions

The current study presents a detailed examination of the long-term pattern of enterococci concentrations in LBG, spanning swimming and non-swimming sites. Despite the absence of enterococci data from 2006–2009, the study elucidated several critical insights. Firstly, it revealed that rain-intensive summers could amplify enterococci levels in LBG, particularly during periods of heightened recreational activity. Users engaging in primary-contact recreation should exercise additional caution and consult relevant advisories before planning such activities in LBG, especially at Weston Park East Beach. From a management perspective, it is recommended that LBG authorities conduct comprehensive sanitary inspections at Weston Park East Beach and assess visitor behaviour across beaches, aiming to identify and promptly control any site-specific factors contributing to pollution. Applying updated and real-time FIB detection (‘nowcast’) technology could provide accurate, timely information on bacteria levels, enabling authorities to respond promptly to changes and mitigate health risks. Furthermore, providing these real-time data to the public could empower recreational users to make informed decisions about their activities in LBG, especially during periods of elevated risk.

Secondly, the effectiveness of ongoing management interventions should be evaluated to understand their contribution towards the observed decrease in faecal contamination in LBG, as indicated by the improved numbers of enterococci exceedances in recent years. These insights can inform the design of more targeted interventions for better water quality management. For example, implementing source tracking methods for enterococci in LBG would be a powerful tool for managing the water quality of this significant body of water. This targeted approach would facilitate the prevention of enterococci contamination at its source, rather than attempting to control it once it has entered the lake. 

Thirdly, a comparison of data collected during non-swimming and swimming seasons and the documentation of visitor numbers and demographic characteristics and uses could offer a more detailed understanding of the influence that recreational users may have on enterococci concentration patterns across the sites. However, collecting such detailed data may require significant resources and time. More importantly, accurately tracking the behaviour of over 30 lake-user groups, as well as visitor numbers, could pose challenges due to the large area of LBG. In addition, initiating a pilot program to assess variations throughout the week or at different times of the day could yield valuable insights into bacterial dynamics and enhance the robustness of the data for analysis. This study also recommends further analyses, considering site-specific characteristics, such as inflows and turbidity, alongside broader climate and land-use influences and in-lake physicochemical parameters to work towards building a hydro-meteorological predictive model for public health. Finally, the observed spatio-temporal variation in the enterococci in LBG underscores the need for a catchment-wide approach to preserve the ecosystem’s health and ensure the safety of the lake for recreational use.

## Figures and Tables

**Figure 1 ijerph-21-00579-f001:**
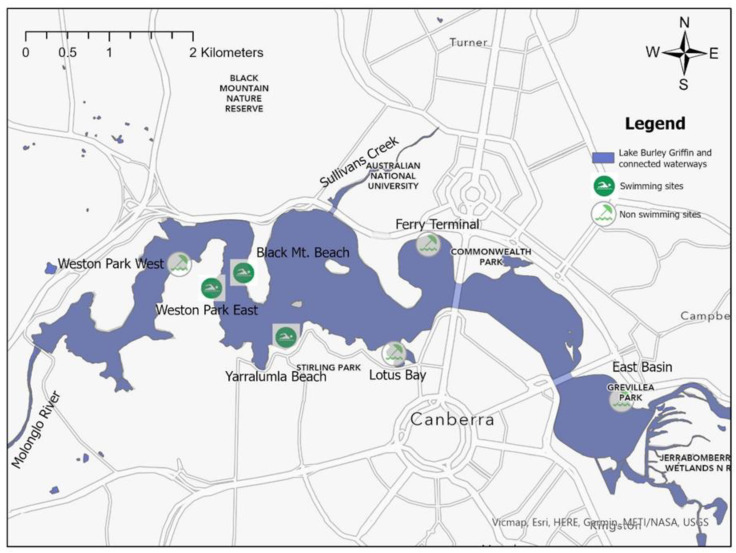
Study area of Lake Burley Griffin and selected recreational sites. Map was created using ArcGIS Pro software (version 3.0.2, Esri, Redlands, CA, USA) [54].

**Figure 2 ijerph-21-00579-f002:**
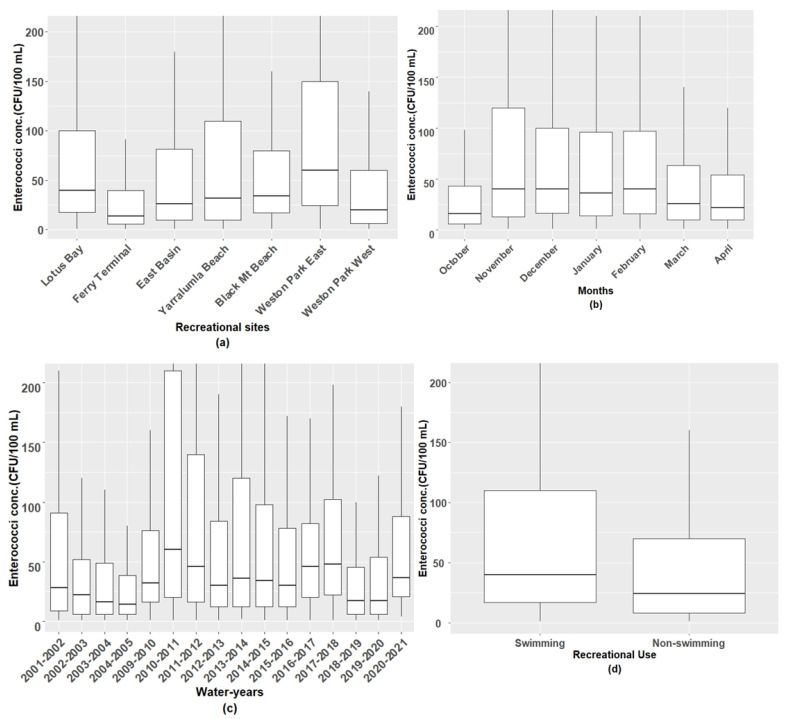
Distribution of enterococci concentrations by percentile for (**a**) recreational sites at Lake Burley Griffin (LBG), (**b**) recreational summer months, (**c**) water-years, and (**d**) primary use of sites. Boxplots depict the variability in enterococci concentrations across samples, with horizontal lines representing the 25th, 50th (median), and 75th percentiles. The box extends from the lower to upper quartiles, with whiskers indicating the range of values, excluding outliers.

**Figure 3 ijerph-21-00579-f003:**
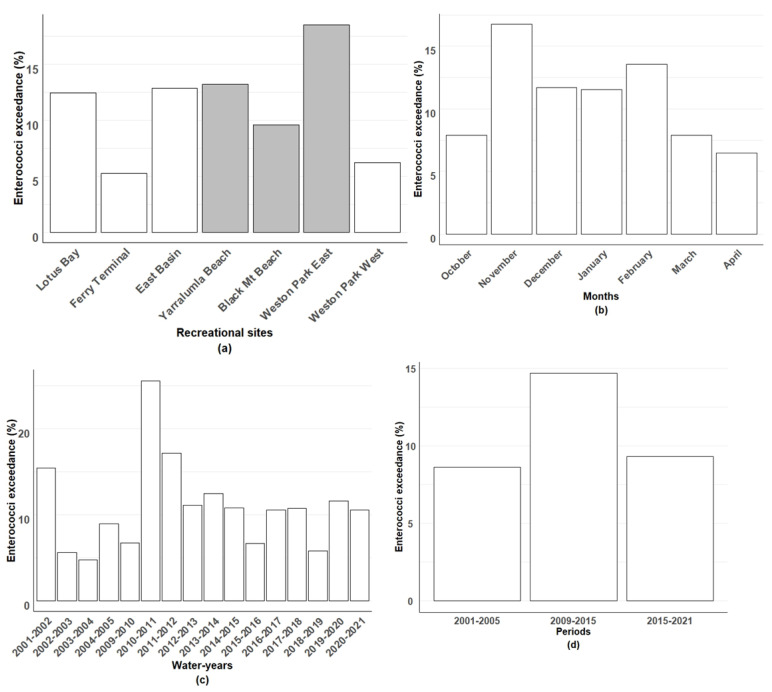
The percentage of enterococci exceedance (above 200 CFU per 100 mL) in weekly water quality monitoring samples for (**a**) recreational sites at Lake Burley Griffin (grey bars indicate swimming sites), (**b**) recreational summer months, (**c**) water-years, and (**d**) three periods from 2001 to 2021.

**Figure 4 ijerph-21-00579-f004:**
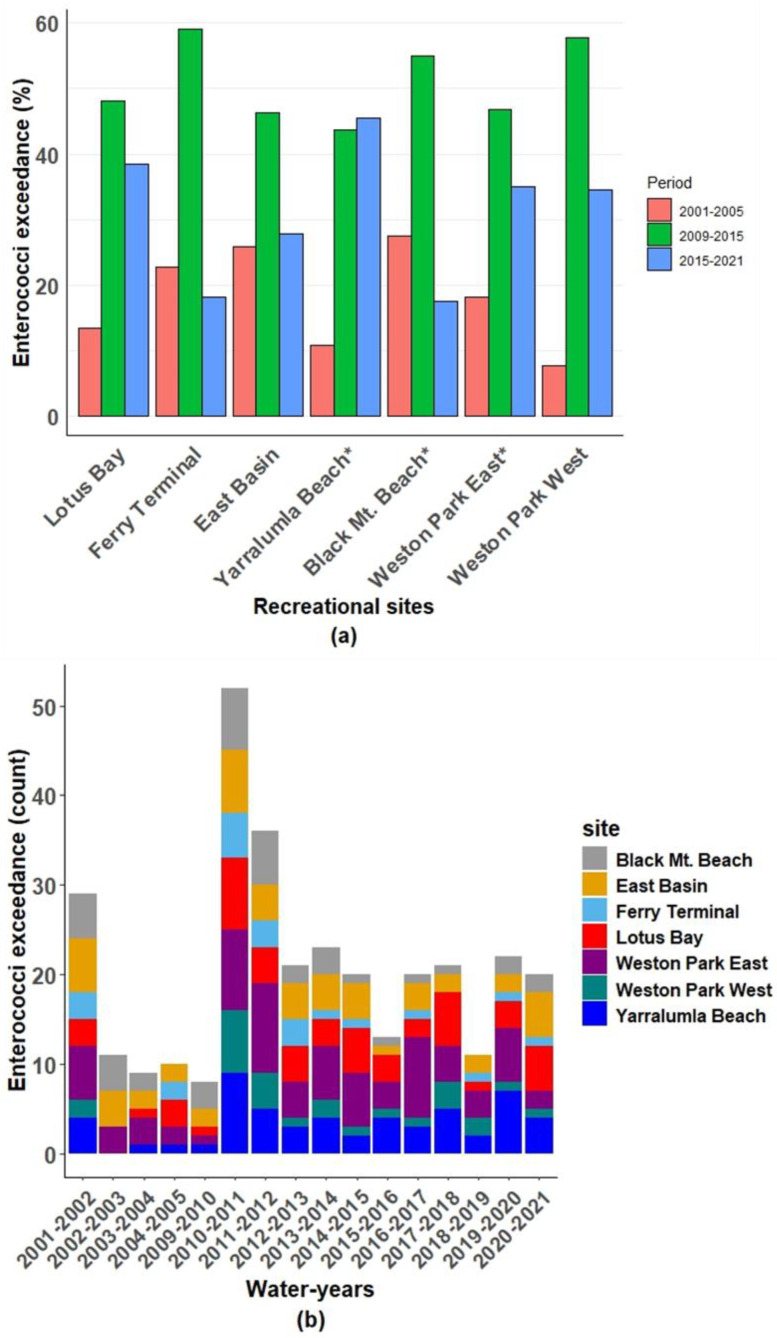
Number of enterococci exceedances (above 200 CFU per 100 mL) across (**a**) recreational sites and periods (* indicates the swimming beaches) and (**b**) recreational sites and months.

**Table 1 ijerph-21-00579-t001:** Study locations, key features, and data coverage.

Site Code	Site Name	LBG Area	Key Features	No. of Years	No. of Samples
LBG510	Lotus Bay	West Lake	Kayak and stand-up paddle board terminal	16	419
LBG511	Ferry Terminal	West Basin	Paddle steamer docking	16	418
LBG512	East Basin	East Basin	Rowing, windsurfing, dragon boating, stand-up paddle boarding	16	420
LBG514	Yarralumla Beach	West Lake	Designated swimming beach, kayak and stand-up paddle board terminal, rowing club	16	416
LBG515	Black Mountain Beach	Tarcoola Reach	Designated swimming beach, rowing, paddle crafting	16	417
LBG516	Weston Park East	Tarcoola Reach	Designated swimming beach, rowing, paddle crafting	16	416
LBG517	Weston Park West	Yarramundi Reach	Designated rowing lanes	16	418

**Table 2 ijerph-21-00579-t002:** Multivariable analysis of spatio-temporal factors associated with exceedance of faecal indicator bacteria (CFU per 100 mL > 200). Odds ratios (ORs) and 95% confidence intervals (CIs) are presented for each variable. Results are based on water quality monitoring data (2001–2021) for Lake Burley Griffin.

Independent Variables	Level (s)	Logit Model			*p*-Value
		OR	95% CI		
			Lower Limit	Higher Limit	
Time-periods		0.98	0.96	1.01	0.13
Months	October	1.00			
	November	2.38	1.54	3.77	<0.001
	December	1.61	1.02	2.58	0.045
	January	1.55	0.98	2.49	0.065
	February	1.76	1.11	2.84	0.018
	March	0.97	0.59	1.62	0.911
	April	0.79	0.42	1.45	0.454
Sites	East Basin	1.00			
	Lotus Bay	0.92	0.61	1.39	0.688
	Ferry Terminal	0.39	0.23	0.64	<0.001
	Yarralumla Beach	1.01	0.67	1.52	0.961
	Black Mountain Beach	0.73	0.47	1.13	0.161
	Weston Park East	1.52	1.04	2.24	0.031
	Weston Park West	0.46	0.28	0.74	0.002

## Data Availability

The water quality data used in this paper are available upon request from the National Capital Authority, Australia (e-mail: lakeburleygriffin@nca.gov.au, website: https://www.nca.gov.au/environment/lake-burley-griffin/water-quality, accessed on 30 June 2021).

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
