# Peer review of "Spatio-Temporal Variation in the Exceedance of Enterococci in Lake Burley Griffin: An Analysis of 16 Years’ Recreational Water Quality Monitoring Data"

_ijerph, 2024, doi:10.3390/ijerph21050579_

Round 1

Reviewer 1 Report (Previous Reviewer 2)

Comments and Suggestions for Authors

Comments follow throughout the attached document.

Author Response

Dear Reviewer, Please see the attached responses based on your valuable comments and feedback.

Reviewer 2 Report (Previous Reviewer 3)

Comments and Suggestions for Authors

The authors complied with the requests.

Author Response

Reviewer 2:

Feedback R2: The authors complied with the requests.

Response R2: We sincerely appreciate the acknowledgement from you regarding our efforts on the manuscript.

Reviewer 3 Report (New Reviewer)

Comments and Suggestions for Authors

1-     Lines 76-78, It's better to picture out such fundamental source of pollution via new figure.

2-     Line 91, Located this 7 monitoring stations in the figure1.  

3-     For better understanding of the result, its suggested to make a new figure with the base of Table.2 data.

4-      Its strongly suggested to transfer result table to supplementary information section and provide new interesting figure instead of table in the manuscript.

5-     Line 304, the mean monthly runoff and discharge graph should be provided in support of such statement.

6-     I think the discussion should be support with appropriate data about land use pattern, waste base location and time series analysis.

7-      Line 316, this hypothesis need to be prove with statistical approach. for example, simple correlation between park visitors in different month vs quality indices

8-     Compare the result of this study in the context of similar regional and globally research work.

9-     How is the collection of quality network in Australia and why this information is not continuing for a significant period of time in the study area?

10- What's the COVID19 pandemic and its restriction on this indices?

11- Several extreme concentration is reported by the authors. what factors make a such situation?

12- What's the innovation of this study? Which items in your research work may be useful for global researcher?

Comments on the Quality of English Language

the text was so fluent and may need minor modification

Author Response

Dear Reviewer, please see the attached responses based on your valuable comments and feedback.

Reviewer 4 Report (New Reviewer)

Comments and Suggestions for Authors

The authors have written an informative paper on faecal bacteria levels in a recreational freshwater lake. The information provided from the monitoring data is informative for future modelling studies etc. in ecological and epidemiological modelling, but there is no new science per se in the paper. Nonetheless, the data gathered and analysed is worthy of publication and I only have a few minor comments which need consideration by the authors before the paper can be published.

My comments are listed below:

(i) There is not much in the paper about the future impact of these results and I suggest that another sentence or two is added in the abstract and conclusions about the impact of these data and the analysis on the likely future response of the lake to faecal bacteria inputs and how the lake could be best managed in the future.

(ii) Page 2, lines 50-54, there is another key mechanism of faecal bacteria into large water bodies and that is via adsorption and desorption to the sediments. An example of this is given in the following paper and I would suggest this is added as it is an MDPI paper: Huang, G., Falconer, R.A., Lin, B. 2015. Integrated river and coastal flow, sediment and escherichia coli modelling for bathing water quality. Water, 7(9), 4752-4777.

(iii) In the introduction there is an important paper widely cited on disease risks in fresh recreational waters and I think this paper should be cited: Wiedenmann, A., et al. 2006. A randomized controlled trial assessing infectious disease risks from bathing in fresh recreational waters in relation to the concentration of Esherichia coli, intestinal enterococci, clostridium, and somatic coliphages. Environmental Health Perspectives, 114(2), February, 228-.

(iv) Another key international figure in this field is David Kay and his paper in this field is widely cited, linking faecal bacteria to health risk and particularly gastroenteritis. Probably the best paper to cite is: Kay, D., et al. 1994. Predicting likelihood of gastroenteritis from sea bathing: results from randomised exposure. The Lancet, 344, October. 905-. 

(v) Page 4, Lines 116-120. Weekly sampling of bacteria is not that frequent for research studies and it would be useful to know the differences at sites for day and night conditions. I appreciate you cannot go back and repeat the data collection periods more frequently, but you could comment that more frequent sampling would be preferable, and suggest this for future studies in the conclusions.

(vi) Page 11, Lines 358-363. I am surprised that faecal bacteria levels were not recorded in key streams discharging into the lake. As for the above comment you should cover this concern and add it to future studies.

(vii) Page 11, Lines 386-388. You mention contributions to contamination in the upstream water catchment. Does this include animals on land? If so then you should say so and suggest fencing off animals from having access to stream in upstream catchments.   

Author Response

Dear Reviewer, please see the attached responses based on your valuable comments and feedback.

Round 2

Reviewer 1 Report (Previous Reviewer 2)

Comments and Suggestions for Authors

Dear,

The manuscript was greatly improved, however, and as previously mentioned, I maintain the same opinion: the scientific content seems very poor, as it was only determined the presence of a single parameter (enterococci) to evaluate the microbiological quality of recreational water.

Author Response

Reviewer 3 Report (New Reviewer)

Comments and Suggestions for Authors

The source of pollution and the inlet dischrage is so important in each quality monitoring program. So I still believe it's essential to have the inlet and outlet statues in the revised manuscript. more over the providing response regarding question #7 is not convincing. If you have any kind of limitation in doing such research, its suggested to mentioned such limitation in the manuscript.

The provided answer to question #10 is not convincing. Why this issue is out of your research? Please focus on important aspect of this event and potential effect of this pandemic on your research.

Comments on the Quality of English Language

Minor English editing may be required

Author Response

This manuscript is a resubmission of an earlier submission. The following is a list of the peer review reports and author responses from that submission.

Round 1

Reviewer 1 Report

Comments and Suggestions for Authors

the draft titled "Spatio-temporal variation of faecal indicator bacteria in Lake Burley Griffin: an analysis of 16 years’ recreational water quality monitoring data" by Ripon Kumar Adhikary et al. collected samples of Lake Burley Griffin in different time (both in year and month) at various sites to investigate the Enterococci exceedance rates, and finally they provided a spatio-temporal trend of FIB in the targeted lake area. Overall, the study is very meaningful, especially for the long term data, which are very valuable. Although the work is not perfect, it deserves publication, and the content is very well fits the scope of the journal.

Major issues:

1. in the titles, the authors mention FIB, while in their study one FIB was investigated, why not change FIB to Enterococci in the title. BTW, Enterococci should be italized.

2. better if the authors provide the water quality data and try to establish the correlation between water quality and FIB exceedance data.

3. data analysis is bit descriptive, based on the data obtained so far, more valuable and important results could be obtained, e.g. is the exceedance is significantly higher for swimming sites in summer months in comparison to autumn months, is one site exceedance significantly higher than the other sites? is there any cross talk between months and sites?  is one time frame exceedance significantly higher than the other two time frame data?

4. better provide spatio-temporal lake use data, how many people in total go to the sites for swimming, fishing, boating... if those data were added, and inclusion into the data analysis would be of great helpful in interpreting the results.

Comments on the Quality of English Language

overall, the language is OK, however the draft still needs improvement.

Reviewer 2 Report

Comments and Suggestions for Authors

Comments follow throughout the attached document.

Reviewer 3 Report

Comments and Suggestions for Authors

The study proposes to evaluate the temporal evolution during 16 years (2001 to 2021) of the presence of FIB in swimming and non-swimming areas in Lake Burley Griffin, Canberra, Australia.

The study is well designed and presents a clear and well-defined methodology.

 Basically it is a database survey seeking to compile data on water contamination after monitoring in that period by the NCA.  Since the data were obtained in the year 2021 and based on the results, it is understood that it is important to include in the Discussion of the study, what were the measures taken by the Local Competent Authorities?  Still on the discussion, is it important to inform in the manuscript, why was there no collection in the period from 2006 to 2009?  In my opinion, the conclusions of the study are too simplistic and do not focus on the measures to be taken.  The conclusion of the study should be based on the results and discussion and not cite other sources.  The conclusion reports that the study is a starting point, but this is controversial, since the monitoring began in 2001 and,
therefore, this study is already a general alert to public health and the competent authorities.
In my opinion, the conclusion needs to be rewritten, alerting more and suggesting public policies that are better
addressed to solve the problem.